# Red blood cell homeostasis in children and adults with and without asymptomatic malaria infection in Burkina Faso

Berenger Kaboré[1,2]*, Annelies Post[1], Mike L. T. Berendsen[1,3], Salou Diallo[2], Palpouguini Lompo[2], Karim Derra[2], Eli Rouamba[2], Jan Jacobs[4,5], Halidou Tinto[2,6], Quirijn de Mast[1], Andre J. van der Ven[1]*

1 Department of Internal Medicine, Radboud Center for Infectious Diseases, Radboud University Medical Center, Nijmegen, The Netherlands, 2 IRSS/Clinical Research Unit of Nanoro (CRUN), Nanoro, Burkina Faso, 3 Open Patient Data Explorative Network, Department of Clinical Research, University of Southern Denmark, Odense, Denmark, 4 Department of Clinical Sciences, Institute of Tropical Medicine (ITM), Antwerp, Belgium, 5 Department of Microbiology and Immunology, University of Leuven (KU Leuven), Leuven, Belgium, 6 Centre Muraz, Bobo-Dioulasso, Burkina Faso

* kaboreberenger@gmail.com (BK); andre.vanderven@radboudumc.nl (AJV)

**Data Availability Statement:** All relevant data are within the paper and its Supporting Information files.

## Abstract

Asymptomatic *malaria* infections may affect red blood cell (RBC) homeostasis. Reports indicate a role for chronic hemolysis and splenomegaly, however, the underlying processes are incompletely understood. New hematology analysers provide parameters for a more comprehensive analysis of RBC hemostasis. Complete blood counts were analysed in subjects from all age groups (n = 1118) living in a malaria hyperendemic area and cytokines and iron biomarkers were also measured. Subjects were divided into age groups (<2 years, 2–4, 5–14 and ≥15 years old) and clinical categories (smear-negative healthy subjects, asymptomatic malaria and clinical malaria). We found that hemoglobin levels were similar in smear-negative healthy children and asymptomatic malaria children but significantly lower in clinical malaria with a maximum difference of 2.2 g/dl in children <2 years decreasing to 0.1 g/dl in those aged ≥15 years. Delta-He, presenting different hemoglobinization of reticulocytes and RBC, levels were lower in asymptomatic and clinial malaria, indicating a recent effect of malaria on erythropoiesis. Reticulocyte counts and reticulocyte production index (RPI), indicating the erythropoietic capacity of the bone marrow, were higher in young children with malaria compared to smear-negative subjects. A negative correlation between reticulocyte counts and Hb levels was found in asymptomatic malaria (ρ = -0.32, p<0.001) unlike in clinical malaria (ρ = -0.008, p = 0.92). Free-Hb levels, indicating hemolysis, were only higher in clinical malaria. Phagocytozing monocytes, indicating erythophagocytosis, were highest in clinical malaria, followed by asymptomatic malaria and smear-negative subjects. Circulating cytokines and iron biomarkers (hepcidin, ferritin) showed similar patterns. Pro/anti-inflammatory (IL-6/IL-10) ratio was higher in clinical than asymptomatic malaria. Cytokine production capacity of ex-vivo whole blood stimulation with LPS was lower in children with asymptomatic malaria compared to smear-negative healthy children. Bone marrow response can compensate the increased red blood cell loss in asymptomatic malaria, unlike in clinical malaria, possibly because of limited level and length of inflammation.

**Funding:** This work was supported by unrestricted grant from SYSMEX Europe GmbH to AV that supported full funding of the presented studies. Furthermore, SYSMEX Europe GmbH provided the analyzers and technical assistance for running the analyzers. The funding source was involved in the study design, but data collection, analysis and interpretation as well as preparation of this report were done independently. https://www.sysmex-europe.com

**Competing interests:** All authors report no potential conflicts. This work was supported by SYSMEX Europe GmbH. This does not alter our adherence to PLOS ONE policies on sharing data and materials.

**Trial registration**: Prospective diagnostic study: ClinicalTrials.gov identifier: NCT02669823.

Explorative cross-sectional field study: ClinicalTrials.gov identifier: NCT03176719.

## Introduction

Malaria is a major cause of anaemia in sub-Saharan Africa with a multifactorial aetiology, including hemolysis, dyserythropoiesis and inflammation induced functional iron deficiency [1]. The aetiology of severe anaemia in clinical malaria is often described [2, 3] where destruction of erythrocytes is compounded by suppressed erythropoiesis [4–6], possibly due to cytokine imbalance [3, 7].

In asymptomatic malaria, however, where *Plasmodium falciparum* (*Pf*) infection does not result in signs of illness and hemoglobin (Hb) levels are less severely affected [8–11]. Hemolysis and splenomegaly, driven by the magnitude of the infecting biomass and chronicity of infection, are recognised as important factors for the development of anaemia in asymptomatic malaria [12]. Furthermore, defining malaria-attributable anaemia is difficult, as many prevalent comorbidities may also lead to dyserythropoiesis and inflammation induced functional iron deficiency. The capacity of the bone marrow to respond to decreasing Hb levels can however be assessed by analysing reticulocyte counts. So far, only one study reported reticulocyte numbers in asymptomatic malaria patients however [10]. In this study, semi-immune children between 5–15 years old with asymptomatic malaria, had lower Hb levels, increased reticulocyte numbers and erythropoietin levels as well as TNF-α levels, compared to healthy non-malaria carriers, indicating dyserythropoiesis. Although *Pf* is known to induce a strong pro-inflammatory response and thereby the development of anaemia, malaria can also induce tolerance to subsequent infections or immune challenges which may limit the development of anaemia in asymptomatic malaria [13, 14].

New generation hematology analysers provide new parameters, such as the reticulocyte production index (RPI), immature reticulocyte fraction (IRF), Delta-He and monocyte phenotypes [15], that enable a more comprehensive understanding of hematological changes and better insight into the bone marrow response to anaemia [16]. Reference values are available from the Western population [17] but are lacking from Africa.

The primary aim of present study was to get a more comprehensive understanding of the hematological changes in subjects with asymptomatic malaria in the various age groups. We hypothesized that in asymptomatic malaria, a pro-inflammatory status together with low grade hemolysis will induce anaemia while immune tolerance may have a protective effect on the erythropoiesis. For this study, we used a new generation hematology analyser that provide parameters to monitor erythropoiesis and the results were combined with markers that influence erytropoieis such as iron biomarkers, circulating cytokines and ex-vivo cytokine production capacity of stimulated whole blood. The study was carried out in Burkina Faso, where prevalences of asymptomatic malaria of up to 75% having been described [18].

## Materials and methods

### Study site and design

Studies were conducted from March 2016 till September 2017 at the Clinical Research Unit Nanoro (CRUN) in a hyperendemic area for *Pf* malaria [19]. Subjects were recruited during two studies, allowing the inclusion of control groups, such as asymptomatic subjects that are

malaria smear negative and patients with clinical malaria. In the present analysis, we included patients with clinical malaria from a prospective diagnostic accuracy study enrolling patients of 3 months and older with acute febrile illness, as described elsewhere [20]. The second study was a cross-sectional field study among 1000 randomly selected healthy volunteers aged over 1-year-old from 24 villages of the Nanoro health and demographic surveillance system (HDSS) [21], allowing inclusion of asymptomatic subjects that are malaria smear negative or positive. See also **S1 File**.

Participants of the two studies were stratified into four age groups: three groups of children (>3 months to <2 years, 2–4 years and 5–14 years) and one group of adults (≥15 years). Furthermore, participants were divided according to case definitions into smear-negative healthy subjects, asymptomatic malaria and clinical malaria. Finally, asymptomatic malaria and smear-negative subjects were combined in a group of asymptomatic subjects and compared with the clinical malaria patients.

## Laboratory analyses

Hematology analyses were performed using the Sysmex XN-1000 haematology analyser (Sysmex Corporation, Kobe, Japan), which generated standard full blood counts including reticulocyte counts as well as research parameters (**S1 Table**) [15]. Furthermore, free haemoglobin (Free-Hb) levels were defined as differences between standard Hb measurements (all RBCs are lysed) and optical Hb level (HGB-O) which measures Hb within intact RBCs. Free-Hb levels above the upper reference limit (i.e. typically ≥0.25 g/L in serum or ≥0.13 g/L in plasma) indicate hemolysis, although free-Hb estimates from the Sysmex analyser and values given by Lippi et al [22] may not be directly comparable. The number of phagocyting monocytes (Phago-MONO), as analysed in the hematology analyser, were used to indicate erythrophagocytosis.

Thick and thin blood films for malaria diagnosis were examined according to World Health Organization procedures [23].

Iron biomarkers Ferritin (FERR), hepcidin (HEP) and soluble transferrin receptor (sTfR) were measured in EDTA plasma samples by quantitative sandwich enzyme immunoassay technique. Four circulating cytokines (IL-6, TNF-α, IFN-ϒ and IL-10) were quantified by MAGPIX technology (Luminex Corporation, Austin, Texas, USA) in EDTA plasma, according to manufacturer's instructions. Ex-vivo cytokine production of cytokines (IL-1β, IL-6, IL-10, TNF-α and IFN-ϒ) was measured from the supernatant of heparinised whole blood after stimulation with lipopolysaccharide (LPS) by standard "sandwich-type" ELISA procedure (ThermoScientific™ Pierce™). Samples from the under-five year's group (<5 years) were used because of the highest burden of malarial anaemia in this group.

## Data management and statistical aspects

Data are reported as median and interquartile range (IQR) unless stated otherwise. Medians were compared using Mann-Whitney U test, and proportional differences were assessed by Pearson chi-squared or Fisher exact test where appropriate. Spearman correlation was used to assess the relationship between continuous variables of interest. STATA 14 (Stata Corp., TX, USA) was used for the statistical analysis. The p values were adjusted for multiple comparison using Benjamin Hochberg procedure. After correcting for multiple testing, the cut-off point of 0.05 was set as limit of significance. Furthermore, a factor analysis taking into account all the biomarkers was performed using the appropriate R software package in order to assess a correlation patterns/clustering trends among the markers. For each marker, a value test (v. test)

representing the importance and the correlation direction (defined by: increase = +; decrease = -) of the marker in the corresponding group was computed.

### Ethical considerations

Study protocols were approved by the national ethics committee of Burkina Faso (ref 2016-01-006). The protocol of the diagnostic study was furthermore approved by the institutional review board of Institut de Recherche en Sciences de la Santé (ref A03-2016/CEIRES), the ethical committee of the university hospital of Antwerp (ref 15/47/492) and the review board of the Institute of Tropical Medicine Antwerp (ref 1029/15). Written informed consent was obtained from all participants or their parents/legal guardians. Assent was obtained from all participants aged 7 to 20 years according to the local requirement.

## Results

### Study population and characteristics

A total of 1118 subjects were enrolled: 191 from the diagnostic accuracy study and 927 from the cross-sectional study (**Fig 1**). **Table 1** describes the demographic and clinical characteristics. The mean (standard deviation) of age were 1.5 (0.4) years, 3.2 (0.9) years, 9.7 (3.0) years, 41.8 (19.7) years in the age category of <2years, 2–4 years, 5–14 years and ≥15 years respectively. In the cross-sectional study, 444 out of 927 (48%) asymptomatic subjects had microscopic positive malaria parasitemia. The proportion of asymptomatic malaria infections in ascending age categories was 29.9%, 38.5%, 69.6% and 30.6% respectively. Parasite density (PD) was significantly higher (p<0.002) in all ages in clinical malaria compared to asymptomatic malaria. *Pf* was the most prevalent species (96.2%).

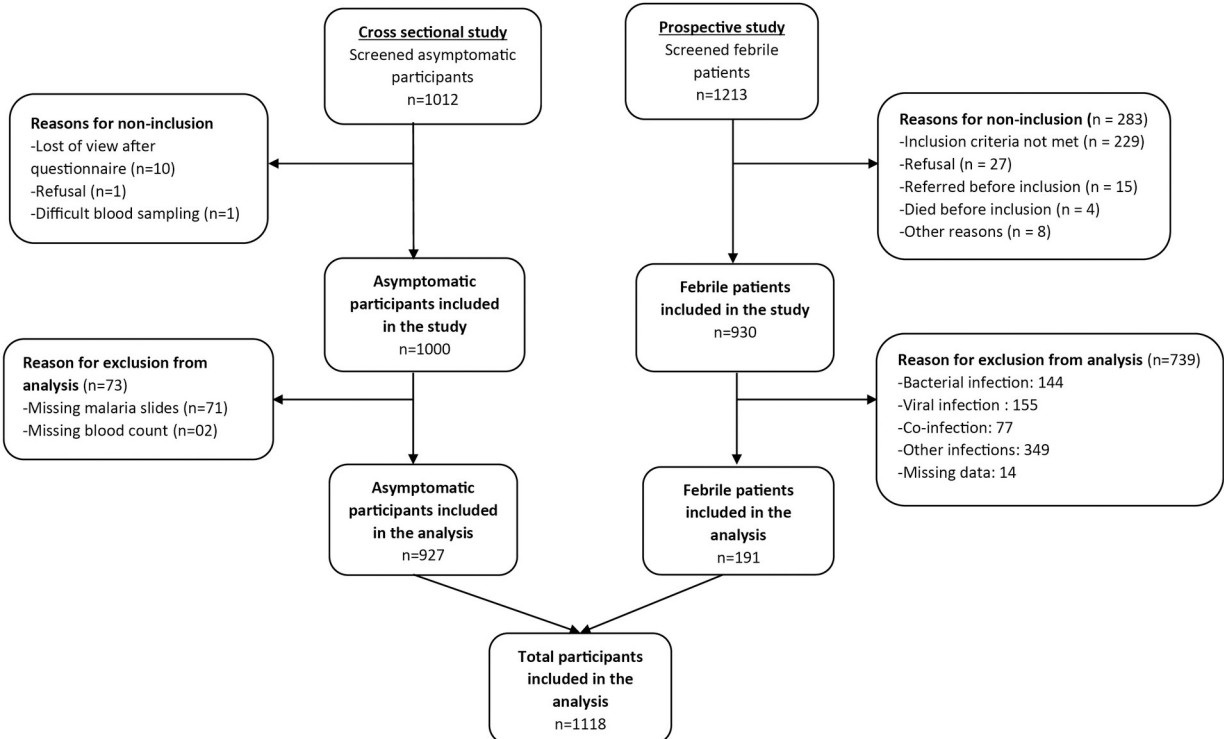

**Fig 1. Flow diagram study subjects.** Diagram represents how the participants included in the analysis were selected from the two studies as described in the methodology section. Only participants with documented malaria result were considered.

**Table 1. Clinical and demographic characteristics.**

| Characteristics | No malaria | Asymptomatic malaria | P value* | Clinical malaria | P value** |
|---|---|---|---|---|---|
| | n = 483 | n = 444 | | N = 191 | |
| Gender | | | | | |
| Female n (%) | 266 (55.1) | 223 (50.2) | | 82 (42.9) | |
| Temperature (˚C) | 36.5 (36.2–37) | 36.5 (36.2–36.9) | NS | 38.6 (38–39.7) | <0.001 |
| Parasite density (/ul) | | | | | |
| less 2yrs | NA | 3405 (515–19418) | NA | 48773 (6167–96104) | |
| 2-4yrs | NA | 4418 (754–16995) | NA | 43183(7254–100613) | <0.002 |
| 5-14yrs | NA | 736 (235–2612) | NA | 6939 (679–27683) | |
| 15yrs and older | NA | 195 (95–645) | NA | 1523 (254–8364) | |
| Anthropometry | | | | | |
| Less 2yrs (number; %) | 96 (19.9) | 41 (9.2) | | 77 (40.3) | |
| Z-wa | -1.2(-1.8.-0.4) | -0.8 (-1.6.-0.1) | 0.06 | NR | |
| Z-ha | -2.2 (-3.1.-1.3) | -1.9 (-2.7.-1.1) | 0.4 | -1.2 (-2.2.-0.3) | <0.006 |
| Z-wh | -0.3 (-1.1. 0.4) | -0.2 (-0.9. 1.0) | 0.3 | NR | |
| MUAC | 15 (14–15) | 15 (14.2–15.5) | 0.9 | ND | |
| 2-4yrs (number; %) | 139 (28.8) | 87 (19.6) | | 66 (34.6) | |
| Z-wa | -1.2 (-1.8.-0.5) | -1.3 (-1.9.-0.5) | 0.5 | NR | |
| Z-ha | -1.8 (-2.6.-1.2) | -2.2(-3.2.-1.2) | 0.7 | -1.5(-2.6.-0.3) | <0.006 |
| Z-wh | -0.3 (-0.9.0.5) | -0.2 (-0.8. 0.6) | 0.2 | NR | |
| MUAC | 15 (14.9–16) | 15.5 (15–16) | 0.2 | ND | |
| 5-14yrs (number; %) | 112 (23.2) | 256 (57.7) | | 28 (14.7) | |
| Z-bmi | 15.0 (14.1–16.2) | 15.2 (14.2–16.5) | 0.6 | NR | |
| MUAC | 17.5 (16–19) | 17.9 (16–19) | 0.7 | ND | |
| 15yrs and older (number; %) | 136 (28.2) | 60 (13.5) | | 20 (10.5) | |
| bmi | 19.8 (18.4–21.9) | 19.0 (18.1–21.4) | 0.4 | | |
| MUAC | 25.3 (24–27) | 25 (23.3–27.9) | 0.9 | ND | |

Data are presented as median and interquartile range unless stated otherwise; **Z-wa**: weight for age Z-score, **Z-ha**: height for age Z-score **Z-wh**: weight for height Z-score and **Z-bmi**: body mass index (kg/m^2) Z-score according to WHO classification; **bmi**: body mass index (kg/m^2); **MUAC**: mid-upper arm circumference; **Yrs**: years; ˚C: degree Celsius; **NS:** not significant; **NA**: not applicable; **ND**: not done; **NR**: not reported. Healthy smear-negative subjects are represented by "No malaria."

*: Comparison between No malaria and Asymptomatic malaria groups

**: Comparison of Clinical malaria group with No malaria and Asymptomatic malaria

Comparison between groups was done by using Mann-Whitney *U* test

## Red blood cell reference values and iron and inflammation biomarkers

First we used the data of the 483 smear-negative healthy subjects to determine reference values (**Table 2, Fig 2** and **S1 Fig**). RBC, as well as absolute and relative reticulocyte counts remained stable with age. In contrast, Hb concentrations and hematocrit (Hct) values increased steadily across the age categories from a median Hb and Hct value in the children <2 years of 9.6 g/dL (IQR: 8.8–10.7) and 31.9% (28.5–33.6) to 12.2 g/dL (11.5–13.4) and 37.4% (34.8–39.9), respectively in those ≥15 years. Age related changes were also seen for other RBC markers, such as MCV, MCH, MCHC, HYPO-HE, HYPER-HE, micro-RBC (MicroR), macro-RBC (MacroR), RET-He, RBC-He (**S1 Fig**).

Although the total number of reticulocytes did not change, their different maturation stages [24] were subject to age-related trends. In addition, both the Hb content of reticulocytes (RET-He) and of RBC (RBC-He) increased across the age categories, resulting in a stable Delta-He (RET-He minus RBC-He) (**S1** and **S2 Figs**). Age specific changes in leukocyte and platelet parameters are given in **S2 Table**.

**Table 2. Red blood cells indices reference values (median, and 5th-95th percentile) per age category in the smear-negative healthy (no malaria) group.**

| Parameters | Age category | | | |
|---|---|---|---|---|
| | Less 2 yrs (n = 96) | 2–4 yrs (n = 139) | 5–14 yrs (n = 112) | 15+ yrs and older (n = 136) |
| **Hb (g/dL)** | 9.6 (7.5–11.9) | 10.8 (8.1–12.4) | 11.6 (9.8–13.2) | 12.2 (10.1–14.7) |
| **HCT (%)** | 31.9 (25.4–36.3) | 34 (26.7–38.6) | 35.3 (30.7–39.8) | 37.4 (30.2–44.1) |
| RBC ($10^6$/μl) | 4.5 (3.5–5.4) | 4.4 (3.5–5.2) | 4.4 (3.5–5.5) | 4.4 (3.5–5.5) |
| **MCV (fL)** | 70.2 (56.2–82.3) | 78.2 (65.5–87.2) | 79.1 (66.9–88.7) | 85.9 (73.0–95.4) |
| **MCH (pg)** | 21.7 (16.7–27.0) | 24.7 (20.0–27.9) | 26.2 (21.6–29.7) | 28.5 (24.0–32.0) |
| **MCHC (g/dL)** | 30.9 (27.4–33.9) | 31.7 (29.0–34.6) | 32.9 (30.5–35.4) | 32.7 (30.8–35.7) |
| **RDW-SD (fL)** | 44.2 (31.5–57.7) | 43.3 (37.1–55.6) | 40.2 (34.9–49.5) | 41.2 (36.5–50.1) |
| **RDW-CV (%)** | 19.1 (14.5–27.4) | 15.8 (12.9–22.5) | 14.1 (12.4–17.1) | 13.2 (12.1–15.5) |
| **HYPO-He (%)** | 10.9 (0.9–64.9) | 1.9 (0.3–25.1) | 0.9 (0.1–8.0) | 0.3 (0.1–2.3) |
| **HYPER-He (%)** | 0.2 (0.0–0.5) | 0.3 (0.1–0.5) | 0.4 (0.2–0.6) | 0.6 (0.3–0.8) |
| **MicroR (%)** | 28.7 (7.1–71.9) | 11.2 (3.0–46.8) | 8.0 (1.4–36.7) | 2.3 (0.5–16.6) |
| **MacroR (%)** | 2.9 (0.8–4.4) | 3.6 (2.0–4.2) | 3.7 (2.4–4.3) | 3.7 (3.0–4.8) |
| RET# ($10^4$/μl) | 5.7 (2.4–14.0) | 6.1 (3.2–23.5) | 6.4 (3.2–20.8) | 5.7 (2.8–12.2) |
| **RET%** | 1.3 (0.5–3.4) | 1.4 (0.7–6.2) | 1.5 (0.7–4.9) | 1.3 (0.7–2.7) |
| HFR# ($10^4$/μl) | 0.2 (0.02–2.2) | 0.1 (0.02–3.2) | 0.09 (0.01–1.5) | 0.07 (0.01–0.7) |
| **HFR%** | 3.1 (0.6–19.0) | 2.4 (0.4–16.6) | 1.5 (0.3–9.7) | 1.2 (0.4–7.4) |
| MFR# ($10^4$/μl) | 0.7 (0.2–2.7) | 0.7 (0.2–3.0) | 0.6 (0.2–2.6) | 0.5 (0.1–1.4) |
| **MFR%** | 11.9 (6.5–17.1) | 10.7 (5.2–17.9) | 9.4 (3.5–17.1) | 8.5 (4.4–14.0) |
| LFR# $10^4$/μl | 4.7 (2.2–10.0) | 5.1 (2.7–18.2) | 5.7 (3.1–15.5) | 5.1 (2.6–10.1) |
| **LFR%** | 83.8 (62.1–92.7) | 86.4 (67.6–93.9) | 89.0 (74.1–96.0) | 90.3 (79.5–95.0) |
| **RPI (%)** | 0.5 (0.2–1.3) | 0.7 (0.3–2.6) | 0.8 (0.4–2.4) | 0.8 (0.4–1.8) |
| IRF# ($10^4$/μl) | 0.9 (0.2–5.0) | 0.8 (0.2–6.0) | 0.6 (0.2–4.1) | 0.6 (0.2–2.0) |
| **IRF%** | 16.2 (7.3–37.9) | 13.6 (6.1–32.4) | 11.1 (4.0–25.9) | 9.8 (5.0–20.5) |
| **RET-He (pg)** | 24.6 (16.8–32.3) | 28.2 (21.0–33.1) | 30.2 (23.7–34.2) | 31.7 (26.0–35.6) |
| **RBC-He (pg)** | 21.7 (15.2–27.1) | 25.4 (19.4–28.4) | 27.0 (21.7–30.2) | 29.2 (23.1–32.4) |
| **Delta-He (pg)** | 3.1 (-0.4–7.6) | 3.0 (-0.2–6.3) | 3.2 (0.4–5.5) | 2.9 (0.0–4.4) |

**Data are presented as** median, and 5th-95th percentiles.

**Yrs:** years; #: absolute count; %: percentage count.

The level of plasma iron biomarkers and inflammatory cytokines changed in the various age groups. Concentrations of sTfR declined with increasing age, with a median (IQR) concentration of 53.8 nmol/L (40.9–69.9) in the children <2 years and 29.5 nmol/L (23.5–35.9) in those ≥15 years. In contrast, plasma ferritin and hepcidin concentrations increased with age from 9.7 (7.8–24.6) ng/ml to 49.6 (22.6–76.4) ng/ml and from 2.0 (0.9–6.2) pg/ml to 11.6 (5.6–21.4) pg/ml, respectively (**Fig 3**). Circulating levels of the proinflammatory cytokine TNF-α decreased across the age categories, while the other cytokines remained stable (IL-6 and IL-10) or were mostly undetectable (IFN-ϒ). (**Fig 5A** and **S3 Table**).

## Different reticulocyte response compensating malarial anaemia in asymptomatic and clinical malaria

Children with clinical malaria had significantly lower Hb levels than both smear-negative subjects and children with asymptomatic malaria. (**Fig 2**). Remarkably, we found no significant differences in Hb levels between the asymptomatic children < 15 years, except those 2–4 years, with or without microscopic parasitemia. The difference in Hb level between asymptomatic subjects and those with clinical malaria was most pronounced in the young children with a

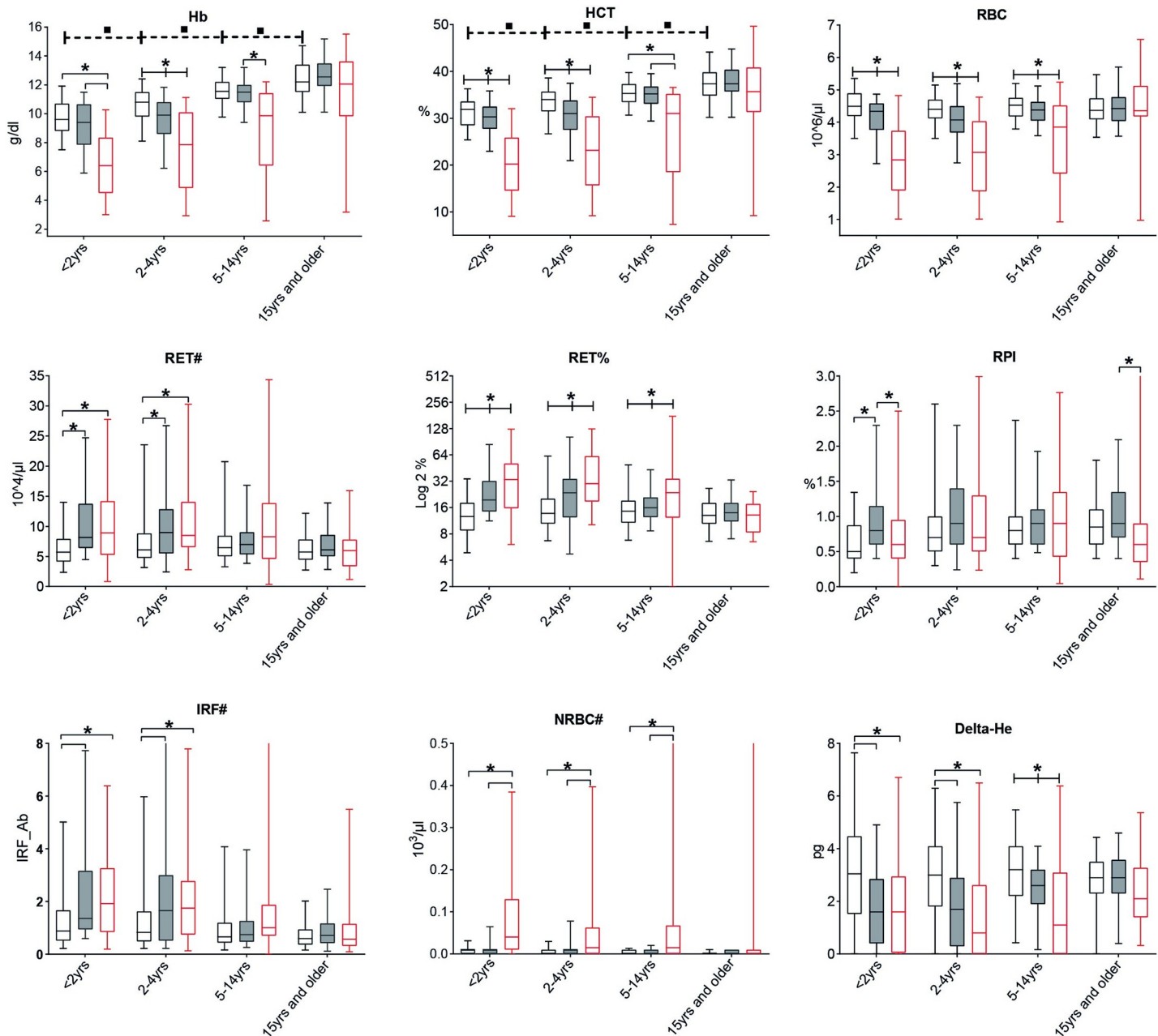

**Fig 2. Hemoglobin (Hb), hematocrit (HCT), red blood cells count (RBC), reticulocyte count (RET#) and percentage (RET%) Reticulocyte Production Index (RPI), Immature Reticulocyte Fraction count (IRF#), nucleated RBC count (NRBC#) and Delta-He per clinical group and age category.** Plots display the status of each hematology parameter per age category. In each age category, participants are divided regarding the health status according to the case definitions whereby healthy smear-negative subjects are represented by "No malaria", smear-positive asymptomatic infections represented by "Asymptomatic malaria" and smear-positive patients with fever are represented by "Clinical malaria". Whiskers bottom and top limits are 5th and 95th percentiles respectively; (): continuous line is used for comparison between clinical status within the same age category and each clinical status was compared with all the other status; (—): dotted line is used for comparison between age category in the "No malaria" group; *: p value with statistically significant difference (p<0.05) between clinical status within the same age category; •: p value with statiscally significant difference (p<0.05) between age category in the "No malaria" group; Mann-Whitney *U* test was used for comparison between groups; yrs: years; #: absolute count; %: percentage count. ☐ No malaria ▨ Asymptomatic malaria ☐ Clinical malaria.

maximum difference (median) in Hb level of 2.2 g/dl in children <2 years decreasing to 0.1 g/dl in those aged ≥15 years (12.1 g/dl versus 12.2 g/dl). Haematocrit levels and RBC counts followed similar trends. In adults, there was no difference between the groups in any of these parameters.

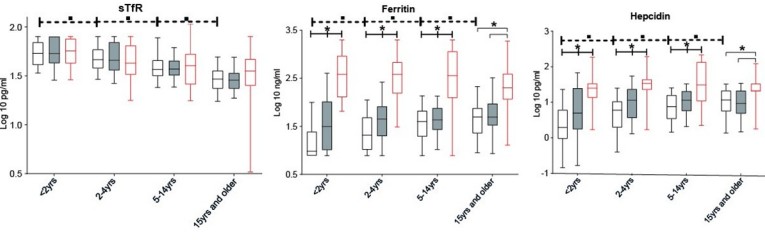

**Fig 3. Iron biomarkers, soluble Transferrin Receptor (sTfR), ferritin and hepcidin per clinical group and age category.** Plots display the level of iron biomarker per age category. In each age category, participants are divided regarding the health status according to the case definitions whereby healthy smear-negative subjects are represented by "No malaria", smear-positive asymptomatic infections represented by "Asymptomatic malaria" and smear-positive patients with fever are represented by "Clinical malaria". Whiskers bottom and top limits are 5 th and 95 th percentiles respectively; (): continuous line is used for comparison between clinical status within the same age category and each clinical status was compared with all the other status; (—): dotted line is used for comparison between age category in the "No malaria" group; *: p value with statistically significant difference (p<0.05) between clinical status within the same age category; •: p value with statistically significant difference (p<0.05) between age category in the "No malaria" group; Mann-Whitney *U* test was used for comparison between groups; yrs: years. ▭No malaria ▭ Asymptomatic malaria ▭ Clinical malaria.

Both absolute and relative reticulocyte counts, estimators of bone marrow activity, were elevated among clinical and asymptomatic malaria infections compared to smear-negative subjects (**Fig 2**). These differences (in particular for the relative count) were only observed in children in the age categories up to 15 years of age. Delta-He is low in acute inflammation but not in chronic inflammation [25]. Delta-he levels are low, both in clinical and asymptomatic malaria in children, indicating a recent effect of malaria on erythropoiesis (**Fig 2**). Adequacy of the bone marrow response for the level of anaemia was analyzed using the reticulocyte production index (RPI). RPI was elevated in children (with significant difference in the <2 years category) with asymptomatic malaria, unlike clinical malaria patients, compared to smear-negative healthy subjects (**Fig 2**). This was supported by the finding of similar reticulocyte numbers and a similar immature reticulocyte fraction (IRF), an early marker of erythroid regeneration, among asymptomatic malaria and clinical malaria in children despite different Hb levels. Moreover, we observed a significant negative correlation between absolute reticulocyte counts and Hb levels in asymptomatic malaria (ρ = -0.32, p<0.001) while in clinical malaria, a nearly flat correlation line was found (ρ = -0.008, p = 0.92) (**Fig 4**). Furthermore, we found that the percentage of immature reticulocytes (HFR and MFR) increase while the opposite is observed for the mature reticulocytes (LFR). This effect is seen in children < 5 years old with no differences between clinical and asymptomatic malaria (**S2 Fig**). Finally, the number of nucleated RBC (NRBC) numbers, that are only seen in case of extreme erythropoietic activity, were only increased in children with clinical malaria (**Fig 2**).

## Different circulating cytokine levels and ex-vivo cytokine production capacity in young children with or without asymptomatic malaria

Inflammation may interfere with erythrocyte homeostasis in different ways [26]. Levels of circulating pro- (IL-6, TNF-α and IFN- ϒ) and anti-inflammatory cytokines (IL-10) were highest in clinical malaria. Nevertheless, cytokine levels were also elevated in asymptomatic malaria compared to smear-negative subjects (**Fig 5A**). In both malaria groups, IL-6 levels correlated significantly with parasite density (ρ = 0.49 and ρ = 0.48 respectively, p<0.001) (**Fig 5B**). The

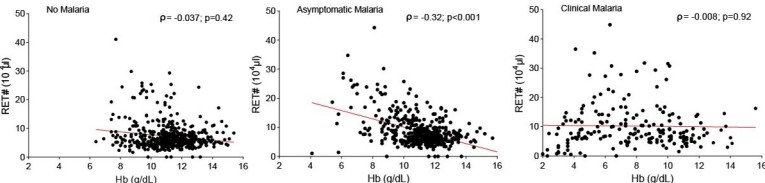

**Fig 4. Correlation between hemoglobin (Hb) and reticulocytes absolute count (RET#) in no malaria, asymptomatic malaria and clinical malaria cases.** Correlation between the hemoglobin level and reticulocytes absolute count in three groups classified according to the health status and based to the case definitions whereby healthy smear-negative subjects are represented by "No malaria", smear-positive asymptomatic infections represented by "Asymptomatic malaria" and smear-positive patients with fever are represented by "Clinical malaria". Spearman test was used to assess the correlation. The line represents the nonlinear fit regression line between the variables.

same trend was seen for TNF-α, IFN-γ and IL-10 (data not shown). Apart from lower levels of circulating proinflammation markers because of lower levels of parasitemia (**Table 1**), immune tolerance may also play a role in asymptomatic malaria. The IL-6/IL-10 ratio, indicating pro/anti-inflammatory balance, was significantly lower in asymptomatic children, compared to clinical malaria (**Fig 5A**). In addition, ex-vivo cytokine production after LPS stimulation was significantly lower for all cytokines, except for IL-6 (p = 0.053), in the asymptomatic malaria infections below the age of 5 years compared to smear-negative healthy subjects (**Fig 5C**).

## Erythrophagocytosis is increased in asymptomatic malaria, especially in young children

Apart from bone marrow suppression, inflammation may lead to anaemia by reducing survival time of erythrocytes by activating macrophages and increase their phagocytozing capacity. We found that the number of phagocytozing monocytes (Phago-MONO) and activated monocytes (RE-MONO) was increased in both asymptomatic and clinical malaria compared to smear-negative healthy subjects in all age categories, the values being highest for clinical malaria. The differences were most striking in children <5 years of age (**Fig 6A**). We found no relation between phagocytozing monocytes numbers and inflammation, nor Hb levels. In contrast, the number of activated monocytes, that followed the same trend as phagocytizing monocytes (**Fig 6A**) related well to Hb levels (**Fig 6B**).

Next to erythrophagocytosis, reduced survival time of erythrocytes can be caused by hemolysis. Mean (standard deviation) free-Hb levels were 2.4 (4.2) g/L in clinical malaria which is significantly higher than 0.8 (3.4) g/L in asymptomatic malaria infections or 0.9 (3.4) g/L in smear-negative healthy subjects (**Fig 7A**). Remarkably, free-Hb levels were related to level of parasitemia in asymptomatic malaria infections, but not in clinical malaria (**Fig 7B**).

## Iron and inflammation is altered in asymptomatic malaria, especially in young children

Inflammation may also influence iron homeostasis. IL-6 is an important driver of hepcidin and ferritin levels [27]. **Fig 3** shows the iron biomarkers in the different age categories and health status. In line with the inflammation data, ferritin and hepcidin levels are high in clinical malaria, followed by asymptomatic malaria and smear-negative healthy subjects in all age categories. At the same time, ferritin and hepcidin levels increase with age in the asymptomatic malaria and smear-negative healthy subjects, while sTfR, that was not different according to the health status, decreases. In contrast, there seems to be no clear age-relation for hepcidin and ferritin in patients with clinical malaria.

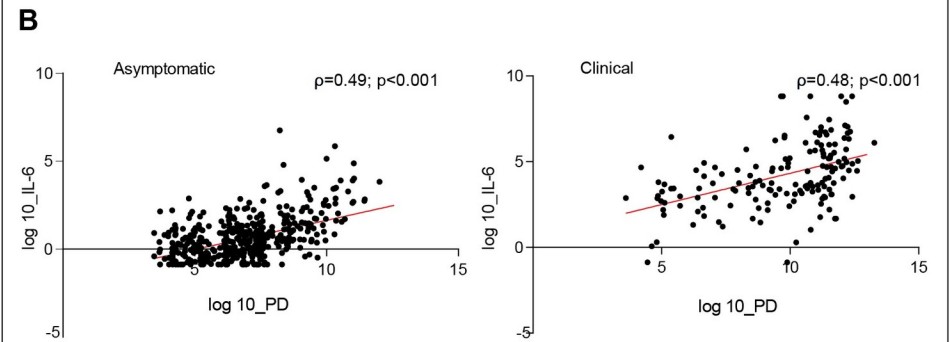

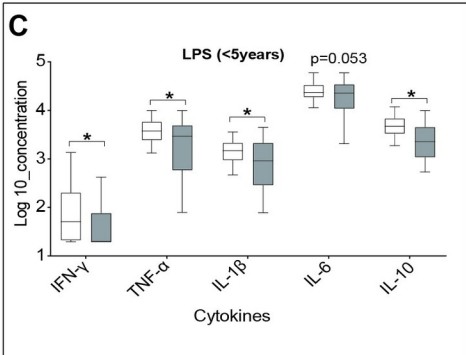

**Fig 5.** A. Circulating cytokines levels (IL-6, IFN-ϒ, IL-10 and TNF-α) per clinical group and age category and Ratio IL-6/IL-10 in malaria infected subjects according to age groups. In each age category, participants are divided regarding the health status according to the case definitions whereby healthy smear-negative subjects are represented by "No malaria", smear-positive asymptomatic infections represented by "Asymptomatic malaria" and smear-positive patients with fever are represented by "Clinical malaria". Whiskers bottom and top limits are 5th and 95th percentiles respectively; (): continuous line is used for comparison between clinical status within the same age category and each clinical status was compared with all the other status; (—): dotted line is used for comparison between age category in the "No malaria" group; *: p value with statistically significant difference (p<0.05) between clinical status within the same age category; •: p value with statiscally significant difference (p<0.05) between age category in the "No malaria" group; Mann-Whitney *U* test was used for comparison between groups; yrs: years. ⬜No malaria ▨ Asymptomatic malaria ⬜Clinical malaria. B. Correlation between IL-6 levels and parasite density (PD) in asymptomatic malaria and clinical malaria cases as previously classified. Spearman test was used to assess the correlation. The line represents the nonlinear fit regression line between the variables. C. Ex-vivo cytokines production (IFN-ϒ, TNF-α, IL-1β, IL-6 and IL-10) after whole blood stimulation with Lipopolysaccharide (LPS) in asymptomatic children less than 5 years old, that were microscopically malaria positive or negative. Whiskers bottom and top limits are 5th and 95th percentiles respectively; (): continuous line is used for comparison between clinical status *: p value with statistically significant difference (p<0.05) between clinical groups; Mann-Whitney *U* test was used for comparison between groups. ⬜No malaria ▨ Asymptomatic malaria.

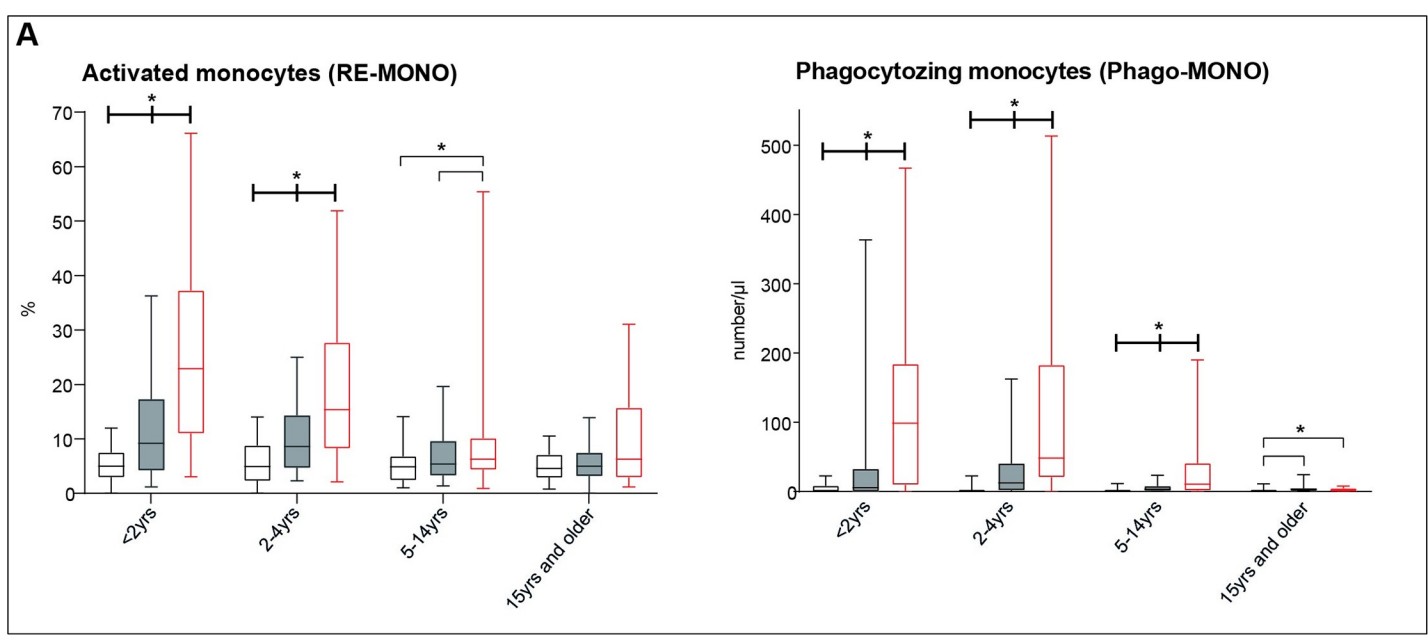

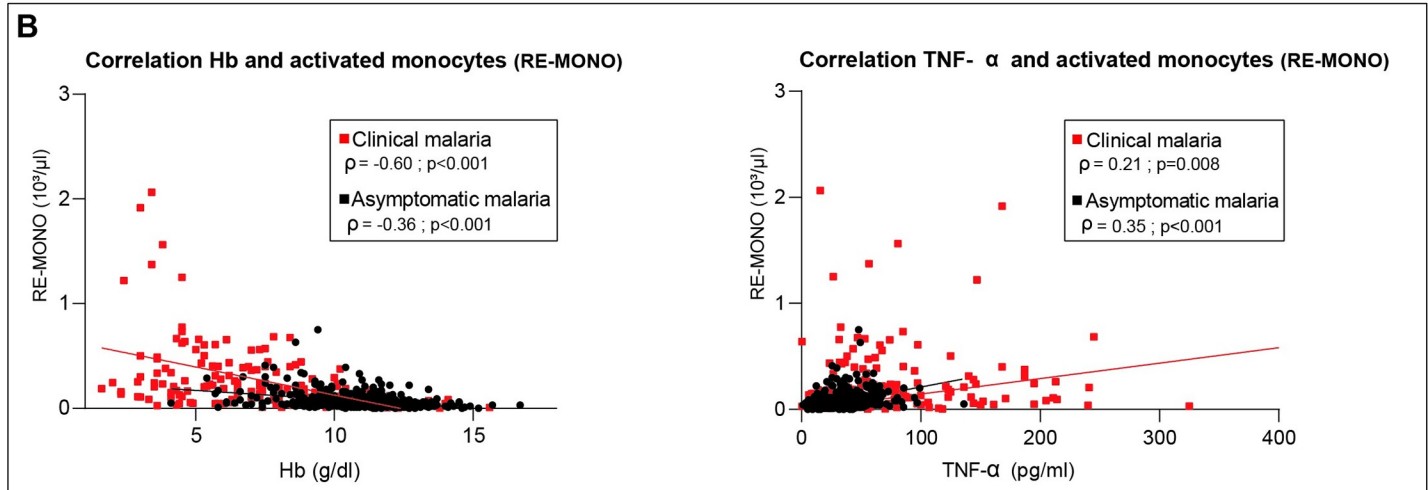

**Fig 6.** A. The proportion of activated monocytes (RE-MONO%) over total number of monocytes and phagocytozing monocytes count (Phago-MONO) per clinical status and age category. In each age category, participants are divided regarding the health status according to the case definitions whereby healthy smear-negative subjects are represented by "No malaria", smear-positive asymptomatic infections represented by "Asymptomatic malaria" and smear-positive patiens with fever are represented by "Clinical malaria". Whiskers bottom and top limits are 5 th and 95 th percentiles respectively; (): continuous line is used for comparison between clinical status within the same age category and each clinical status was compared with all the other status; *: p value with statistically significant difference (p<0.05); yrs: years; Mann-Whitney U test was used for comparison between groups. ☐ No malaria ▨ Asymptomatic malaria ☐ Clinical malaria. B. Correlation between Activated Monocytes count (RE-MONO#), haemoglobin (Hb) and ciculating TNF-α levels in "Asymptomatic malaria" and "Clinical malaria" as previously classified. Spearman test was used to assess the correlation. The lines represents the nonlinear fit regression line between the variables for each clinical status and as indicated in the legend (Red for "Clinical malaria" and Black for "Asymptomatic malaria").

### The biomarkers are clustered according to malaria status

A factor analysis including all the biomarkers indicate a clustering trend of the biomarkers according to the participants health status as displayed on the **S3 Fig,** The extend (increase or decrease) in which each biomarkers is expressed depending on the health status is presented in the **S4 Table.**

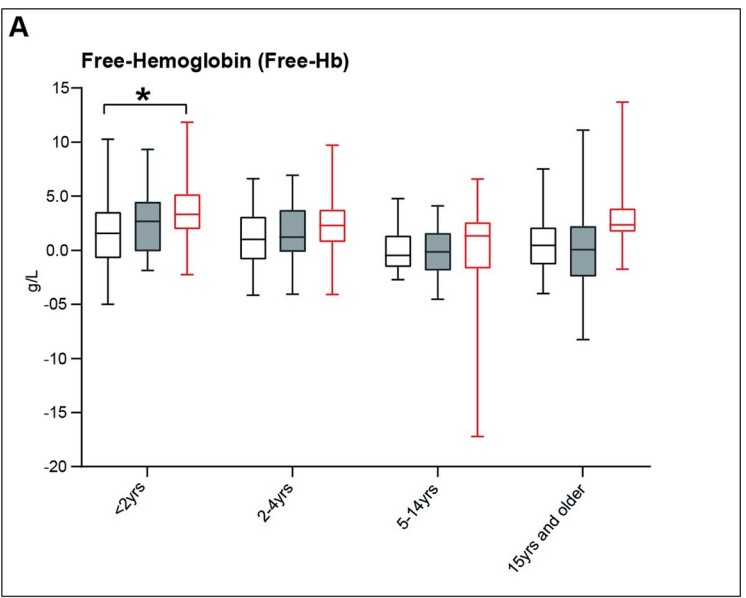

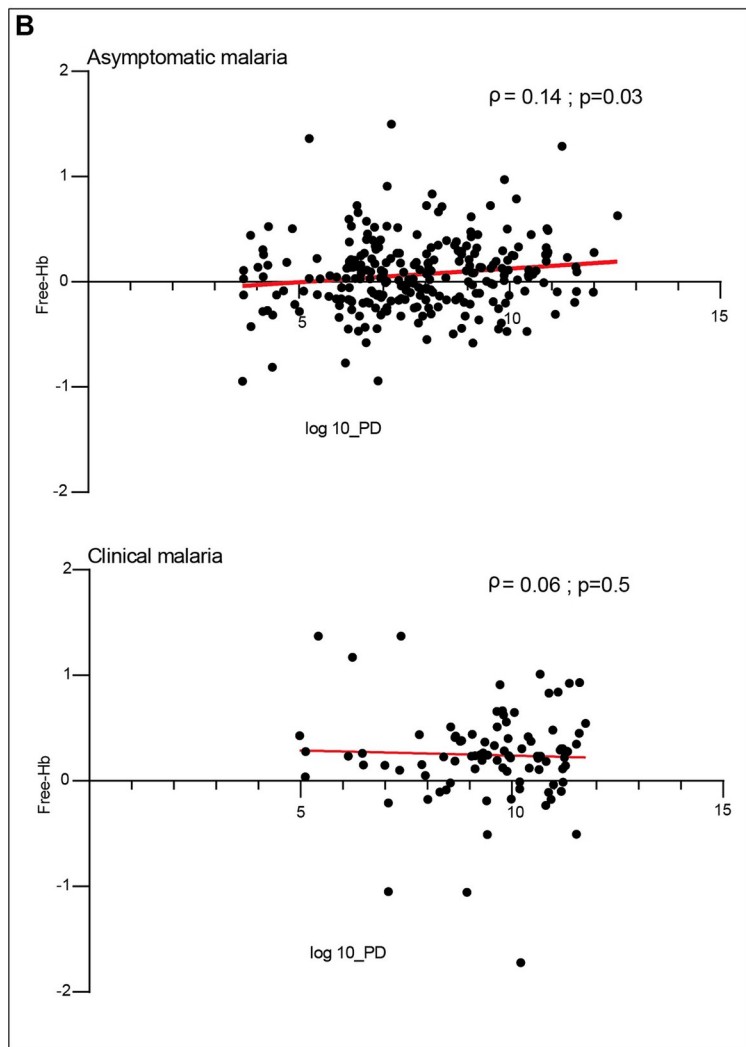

**Fig 7.** A. Free-hemoglobin levels (g/L) per clinical status and age category. In each age category, participants are divided regarding the health status according to the case definitions whereby healthy smear-negative subjects are represented by "No malaria", smear-positive asymptomatic infections represented by "Asymptomatic malaria" and smear-positive patienst with fever are represented by "Clinical malaria". Whiskers bottom and top limits are 5 [th] and 95 [th] percentiles respectively; (): continuous line is used for comparison between clinical status within the same age category and each clinical status was compared with all the other status; *: p value with statistically significant difference (p<0.05); yrs: years; Mann-Whitney *U* test was used for comparison between groups. ⊏══⊐ No malaria ⊏▨▨⊐ Asymptomatic malaria ⊏══⊐ Clinical malaria. B. The correlation between the free-haemoglobin (Free-Hb) and the parasite density (PD) in "Asymptomatic malaria" and "Clinical malaria" as previously classified. Spearman test was used to assess the correlation. The line represents the nonlinear fit regression line between the variables.

## Discussion

The hematological reference values for people living in a malaria hyperendemic area in Burkina Faso indicate that Hb levels and erythrocyte indices increase with age. The concomitant changing iron and inflammation parameters suggest an age-related improvement of iron incorporation. Hb levels did not differ between asymptomatic malaria and smear-negative healthy subjects, while the presence of higher reticulocyte response in asymptomatic malaria indicate that RBC losses are adequately compensated by bone marrow response. Our data further suggest that the effect of malaria on the bone marrow is mostly acute while a limited pro-inflammatory response in asymptomatic malaria may preserve bone marrow response.

Our study had several strenghts, as the inclusion of more than 1100 participants from different age groups allowed us to provide hematological reference ranges for a malaria hyperendemic area. Control groups were included to compare asymptomatic malaria with smear negative asymptomatic subjects and those with clinical malaria. Furthermore, the combined assessment of hematology parameters, iron biomarkers and circulating and ex-vivo cytokines improved our insight in RBC homeostasis.

Nevertheless, a number of limitations need to be noted. First, malaria classification was based on microscopy therefore sub-microscopic malaria infections may have been included in the smear-negative group. Second, we did not screen for other infections like hookworm or hemoglobinopathies and the ex-vivo cytokines production was only available for the under-five years. For ethical reasons, all smear positive asymptomatic infections were given malaria treatment, so we can not provide evidence for the acute or chronic nature of the infection.

Our finding of similar Hb levels among asymptomatic subjects that are smear positive or negative, is in contrast with previous reports [9, 28–32]. The level of malaria endemicity, co-occuring conditions such as iron deficiency, hemoglobinopathies and intestinal parasites, may explain the found differences. Our study, carried out in a high malaria endemic area with 48% of healthy population being smear positive, shows that asymptomatic malaria infections, unlike clinical malaria patients, have an adequate bone marrow response to decreasing Hb levels, using reticulocyte numbers and RPI as indicators. Previous reports suggest that chronic hemolysis and splenomegaly, driven by the magnitude of the infecting biomass and chronicity of infection, underly the development of anaemia in asymptomatic malaria [12]. Our finding of low Delta-He levels, indicating the difference in hemoglobinization between reticulocytes and erythrocytes, suggests a more acute effect of malaria on the bone marrow however [25]. The hemoglobin content of reticulocytes [reported as RET-H$_e$ on Sysmex analysers (Sysmex Corporation, Kobe, Japan) and as CHr on ADVIA analysers from Siemens (Siemens Medical Solutions Diagnostics, Erlangen, Germany)] is an indicator for iron incorporation in the hem over the previous 2–4 days [33]. The hemoglobin content of mature red blood cells (RBC-H$_e$ on Sysmex analysers and CH on Bayer analysers) reflect iron availability over a much longer

period, as erythrocytes have a lifespan of 100–120 days [33]. Hemolysis may reduce lifespan of RBCs in asymptomatic malaria but the difference with reticulocytes will remain significant.

Free-Hb levels, as measured in our study, were only higher in clinical malaria patients compared to smear negative infections, which argues against a significant role of hemolysis in asymptomatic malaria [12]. However, for acurate estimation, the amount of free-Hb that is bound, sequestred and processed should be known. Furthermore, blood was collected from asymptomatic subjects during a field study and we can not exclude artificial hemolysis due to sample processing, which may obscure differences between those that were smear positive or negative.

Erythropoiesis may be suppressed by increased inflammation. We found increased levels of circulating pro-inflammatory cytokines in asymptomatic and clinical malaria, that related to levels of parasite densities, however, considering also the anti-inflammatory cytokine IL-10, we found that the IL-6/IL-10 ratio was significantly lower in asymptomatic children, compared to clinical malaria, as reported before [34, 35]. This difference in ratio was not noticed in adults, possibly explaining the similar Hb levels in adults with or without clinical malaria. In addition to circulating cytokine measurements, we also assessed ex-vivo whole blood cytokine production after LPS stimulation and found dampened responses in asymptomatic malaria infections as reported [35–37], suggesting immune-tolerance in these subjects. Apart from suppression of erythropoiesis, inflammation may also restrict erythroid cell differentiation and proliferation. We indeed found that NRBCs were prominent in clinical malaria, unlike asymptomatic malaria.

In line with the inflammation data, we found that ferritin as well as hepcidin are high in clinical malaria while only mildly elevated in asymptomatic malaria. Inflammation significantly influences iron homeostasis and both the major iron storage protein ferritin and the major iron transport regulating protein hepcidin are well known acute-phase reactants. A recent study indicated that these iron biomarkers are influenced by acute clinical malaria and during convalescence [38]. On the other hand, levels of sTfR were not different between the various clinical status, which is in contrast to findings from Righetti et al [39]. Evaluation of sTfR levels in our study is complicated as inflammation, malaria, age and iron deficiency all play a role. Looking at the different clinical status in the various age categories, we found increasing Hb levels as subjects get older as well as RBC indices (MCV, MCHC, MCH), increasing HEP and FERR levels combined with decreasing pro-inflammatory cytokines and sTfR levels. Taken together, these data suggest an increased availability of iron for erythropoiesis or increased iron incorporation over time. Also new RBC indices suggest similarly, as MacroR, HYPER-He, RBC-He and RET-He increase while MicroR, HYPO-He decrease.

Apart from hemolysis, iron disturbance and suppression of erythropoietic activity, enhanced erythrophagocytosis may also contribute to anaemia [27]. Phagocytic monocytes (Phago-MONO) are detected on Sysmex hematology analysers as research parameter. The presence of phagocytic monocytes was prominent in clinical malaria, followed by asymptomatic malaria and smear-negative healthy subjects, and mostly only in children <5 years. Importantly, the number of phagocytic monocytes did not correlate with Hb levels nor with circulating cytokine levels, as increased phagocytosis of uninfected RBC is thought to be the main contributor of anaemia in malaria [1, 40, 41]. In contrast, the number of activated monocytes did correlate with Hb levels and circulating cytokine levels.

## Conclusions

Bone marrow response, using reticulocyte numbers, RPI and hemoglobin levels as indicators, seem to compensate the increased red blood cell loss in asymptomatic malaria infections, unlike in patients with clinical malaria. Low Delta-He levels as were found in asymptomatic

and clinial malaria, suggests mostly a more acute effect of malaria. Our data also suggests that limited inflammation in asymptomatic malaria may play a role in the preservation of the bone marrow response to the higher turnover of RBC driven by the magnitude of the infecting biomass and chronicity of infection.

## Supporting information

**S1 Fig. RBC indices (MCV, MCH, MCHC, MicroR, MacroR, HYPO-He, HYPER-He) and hemoglobinization of reticulocytes (RET-he) and RBC (RBC-He) per clinical group and age category.** Plots display the status of each haematology parameter per age category. In each age category, participants are divided regarding the health status according to the case definitions whereby healthy smear-negative subjects are represented by "No malaria", smear-positive asymptomatic cases represented by "Asymptomatic malaria" and smear-positive patienst with fever are represented by "Clinical malaria". Whiskers bottom and top limits are 5th and 95th percentiles respectively; (): continuous line is used for comparison between clinical status within the same age category and each clinical status was compared with all the other status; (—): dotted line is used for comparison between age category in the "No malaria" group; *: p value with statistically significant difference (p<0.05) between clinical status within the same age category; ▪: p value with statiscally significant difference (p<0.05) between age category in the "No malaria" group; Mann-Whitney *U* test was used for the comparison between groups; **NS:** not significant; yrs: years; %: percentage. ⬜ No malaria ⬜ Asymptomatic malaria ⬜ Clinical malaria.
(TIF)

**S2 Fig. High fluorescence reticulocytes (HFR), medium fluorescence reticulocytes (MFR), low fluorescence reticulocytes (LFR) per clinical group and age category.** Plots display the status of each haematology parameter per age category. In each age category, participants are divided regarding the health status according to the case definitions whereby healthy smear-negative subjects are represented by "No malaria", smear-positive asymptomatic cases represented by "Asymptomatic malaria" and smear-positive patienst with fever are represented by "Clinical malaria". Whiskers bottom and top limits are 5th and 95th percentiles respectively; (): continuous line is used for comparison between clinical status within the same age category; (—): dotted line is used for comparison between age category in the "No malaria" group; *: p value with statistically significant difference (p<0.05) between clinical status within the same age category; ▪: p value with statiscally significant difference (p<0.05) between age category in the "No malaria" group; Mann-Whitney U test was used for the comparison of median between groups; **NS:** not significant; yrs: years; %: percentage. ⬜ No malaria ⬜ Asymptomatic malaria ⬜ Clinical malaria.
(TIF)

**S3 Fig. Principal component analysis displaying the clustering of the participants based on their health status in the 3 main health categories.**
(TIF)

**S1 File. Supporting methods.**
(DOCX)

**S1 Database.**
(ZIP)

## Acknowledgments

The authors would like to thank the following people: 1. Nurses from CMA, the laboratory technicians from CRUN, the team of data managers from CRUN, the field team and the study nurses from CRUN–Clement Zongo, Bakombania Abassiri, Catherine Nikiema, Esther Kapioko, Celine Nare, Souleymane Ouedraogo and Alassane Compaore for their dedication to the study, 2. Laboratory technicians Helga Dijkstra and Heidi Lemmers at RadboudUMC laboratory of experimental internal medecine, 3. Sysmex Inc, particularly Dr Marion Munster, Dr Jo Linssen and Dr Jarob Saker for their technical assistance, 4. All study participants and their families.

## Author Contributions

**Conceptualization:** Berenger Kaboré, Annelies Post, Mike L. T. Berendsen, Salou Diallo, Palpouguini Lompo, Karim Derra, Eli Rouamba, Jan Jacobs, Halidou Tinto, Quirijn de Mast, Andre J. van der Ven.

**Data curation:** Berenger Kaboré, Annelies Post, Mike L. T. Berendsen, Salou Diallo, Quirijn de Mast.

**Formal analysis:** Berenger Kaboré, Annelies Post, Mike L. T. Berendsen, Andre J. van der Ven.

**Funding acquisition:** Andre J. van der Ven.

**Investigation:** Berenger Kaboré, Annelies Post, Mike L. T. Berendsen, Salou Diallo, Palpouguini Lompo, Quirijn de Mast.

**Methodology:** Berenger Kaboré, Annelies Post, Mike L. T. Berendsen, Salou Diallo, Palpouguini Lompo, Karim Derra, Eli Rouamba, Jan Jacobs, Halidou Tinto, Quirijn de Mast, Andre J. van der Ven.

**Project administration:** Berenger Kaboré, Annelies Post, Mike L. T. Berendsen, Salou Diallo, Halidou Tinto, Andre J. van der Ven.

**Resources:** Berenger Kaboré, Annelies Post, Jan Jacobs, Halidou Tinto, Andre J. van der Ven.

**Software:** Karim Derra.

**Supervision:** Berenger Kaboré, Salou Diallo, Palpouguini Lompo, Karim Derra, Eli Rouamba, Jan Jacobs, Halidou Tinto, Quirijn de Mast, Andre J. van der Ven.

**Validation:** Berenger Kaboré, Annelies Post, Mike L. T. Berendsen, Quirijn de Mast, Andre J. van der Ven.

**Visualization:** Berenger Kaboré, Annelies Post, Mike L. T. Berendsen, Quirijn de Mast, Andre J. van der Ven.

**Writing – original draft:** Berenger Kaboré.

**Writing – review & editing:** Berenger Kaboré, Annelies Post, Mike L. T. Berendsen, Salou Diallo, Palpouguini Lompo, Karim Derra, Eli Rouamba, Jan Jacobs, Halidou Tinto, Quirijn de Mast, Andre J. van der Ven.

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
