## [Decision Letter · Decision Letter 0]

27 Jul 2020

PONE-D-20-20127

Red blood cell homeostasis in children and adults with and without asymptomatic malaria infection in Burkina Faso

PLOS ONE

Dear Dr. KABORE,

Thank you for submitting your manuscript for review to PLoS ONE. After careful consideration, we feel that your manuscript will likely be suitable for publication if the authors revise it to address relevant points raised by the reviewers.  According to reviewers, there are some specific areas where further improvements would be of substantial benefit to the readers, including data analysis and results. For example, reviewer #1 suggests additional analysis for biomarkers correlations;  Reviewer # 2 suggests reviewing data analysis to reduce statistical dispersion;  Reviewer #3 pointed out that the individual data underlying the findings are not fully available in the manuscript and no repository have been provided. For your guidance, a copy of the reviewers' comments is included below. 

We look forward to receiving your revised manuscript.

Kind regards,

Luzia Helena Carvalho, Ph.D.

Academic Editor

PLOS ONE

Journal Requirements:

2. Please provide additional details regarding participant consent. As your study included minors, please state whether you obtained consent from parents or guardians. If the need for consent was waived by the ethics committee, please include this information.

"This work was supported by unrestricted grant from SYSMEX Europe GmbH to AV that supported full funding of the presented studies. Furthermore, SYSMEX Europe GmbH provided the analyzers and technical assistance for running the analyzers. The funding source was involved in the study design, but data collection, analysis and interpretation as well as preparation of this report were done independently.

https://www.sysmex-europe.com"

We note that you received funding from a commercial source: SYSMEX Europe GmbH

Reviewers' comments:

Reviewer's Responses to Questions

**Comments to the Author**

1. Is the manuscript technically sound, and do the data support the conclusions?

Reviewer #1: Yes

Reviewer #2: Yes

Reviewer #3: Partly

2. Has the statistical analysis been performed appropriately and rigorously? 

Reviewer #1: Yes

Reviewer #2: Yes

Reviewer #3: Yes

3. Have the authors made all data underlying the findings in their manuscript fully available?

Reviewer #1: No

Reviewer #2: Yes

Reviewer #3: No

4. Is the manuscript presented in an intelligible fashion and written in standard English?

Reviewer #1: Yes

Reviewer #2: Yes

Reviewer #3: Yes

5. Review Comments to the Author

Reviewer #1: 1) In Table 1, the percentage of each age group should reflect the proportion of each column instead of each row.

2) There are many biomarkers examined here. It will be great to do some in-depth analysis on the correlations of the biomarkers to cluster and group biomarkers, and to draw some results and conclusion at the hyper level.

Reviewer #2: Red blood cell homeostasis in children and adults with and without asymptomatic malaria

infection in Burkina Faso.

Kaboré B. et al.

This paper describes a large set of hematological parameters in a population living in a malaria hyperendemic area. The novelty of the study lies fundamentally in the integration of reticulocyte and bone marrow related data in symptomatic and asymptomatic subpopulations. The hypothesis is that in asymptomatic malaria immune tolerance may have a protective effect on erythropoiesis, which would compensate the anaemia derived from the pro-inflamatory status and hemolysis. The results support a correlation between the bone marrow response, with higher turnover of RBC, and the development of asymptomatic malaria. Differences were found mostly between children showing clinical malaria and asymptomatic or no malaria subjects.

Overall the study appears well designed, the analytical techniques correctly used and the results convincing, however there are some weaknesses which I recommend to address before publication.

Specific comments:

1. Although the statistical analysis performed yields significant differences among the three main groups analyzed (smear-negative, asymptomatic and clinical malaria) for several parameters, some data display a rather large dispersion (figures 2, 3, 5, 6..), particularly in the adult groups in clinical malaria and some asymptomatic malaria sets. This might be explained if these groups included some subjects displaying characteristics not contemplated in the selection criteria, as a concomitant non malaria health issue, or perhaps the inclusion of pregnant women, which has been reported to display differential response to malaria, leading to heterogeneity in the group. The authors may review the characteristics of the individuals included in each study group and test if the statistical dispersion is reduced by the exclusion or regrouping by these criteria.

2. Participants of the asymptomatic group: the subjects were included in this group based on being smear-positive and healthy at the moment of the test. It is stated also (Supporting methods, study design) that a questionnaire was taken at sample collection, but no information is given if the subjects were questioned for febrile episodes in the previous two weeks to ascertain lack of malaria symptoms. It is possible that some of the asymptomatic subjects were actually either developing symptomatic malaria at an early stage or tested after a malaria episode. Related to the comment #1, this would misrepresent both the symptomatic and asymptomatic groups and add uncertainty to the statistics. Ideally the asymptomatic group should correspond with immune-tolerant individuals. If the authors have such information it would be advisable to be mentioned in the text.

3. The group of subjects �15 years is quite broad, it would be convenient to indicate the average age of this group as blood parameters are most likely affected in what is considered elderly people in endemic areas (>40 years) than in young adults (15-40 years)

Minor comments

“NS” in table 1 and figures S1 and S2 is not defined in the text or legends

Quality of figures: axis labels are difficult to read at the resolution provided in the figures containing large number of plots (fig 2, 5, 7, S1)

Reviewer #3: This manuscript by Kaboré et al describes the hematological alterations shown in asymptomatic or clinical malaria and also investigates the bone marrow response to the loss of RBCs depending on the severity of the infection. The study integrates a wide variety of hematological variables, iron biomarkers and cytokines as a comprehensive approach to understanding how the bone marrow compensates for the condition of anaemia. One of the major highlights of the paper is the demonstration that asymptomatic malaria is associated with enhanced erythropoiesis and immune tolerance compared to those with clinical malaria, probably due to a lower imbalance towards the production of pro-inflammatory factors in individuals with asymptomatic malaria.

In recent years, asymptomatic malaria carriers have become a focus of malaria research, as they are considered the silent reservoir of malaria. The factors that protect these individuals from developing clinical symptoms are still unclear, however, this study provides new insights into the underlying processes of subclinical malaria. In addition, the large and significant number of samples analysed from different ages provides solid and consistent results.

It should be pointed out that the individual data underlying the findings are not fully available in the manuscript and no repository have been provided. Although summary statistics are available, the data points behind means or ranges are not given as stated in the PLOS Data policy.

Most of graphs and statements are clear and well-founded. Nevertheless, there are some points that the authors may review to facilitate the readability and interpretation of the results:

Major comments:

1. As the authors note in the discussion, despite the study is focussed on asymptomatic malaria, the diagnostic method used fails to detect those individuals with submicroscopic malaria, known to be the majority of asymptomatic malaria individuals. In addition, there is no information about other pathologies that may cause hematological alterations such as sickle cell disease. For that reason, smear-negative samples should not be considered reference values, as they may include many submicroscopic malaria and hemoglobin S carriers.

2. Some of the statements do not fit exactly with the results shown in the graphs or are too generalistics. Along the results or discussion the authors describe significant differences between infection groups, however they do not indicate the age groups in which this occurs (e.g. line 270, Fig2, one of the under 15 years groups do not show increased reticulocyte numbers; line 272, Fig2, the authors should indicate that Delta-he levels are low only in children; line 274, Fig2, RPI is only elevated in under-2 children; Line 378, Fig2, the authors claim that erythrocyte indices increase with age, however, the RBC numbers are constant along the age groups). For the benefit of the readers, the authors should be more accurate in their description of the results.

3. Lines 403-405: The authors claim that asymptomatic malaria infections, unlike clinical malaria, have an adequate bone marrow response to decreasing Hb levels, using reticulocyte numbers and RPI as indicators, however, reticulocyte numbers show increased numbers in both subclinical and clinical malaria. It is unclear on what differential results between both groups this statement is based.

4. This reviewer does not see a clear reason why the cytokine production after LPS stimulation is not analysed in clinical malaria patients. In line with the rest of data provided where this group is considered. This analysis is important to provide insights into subclinical malaria immune tolerance in comparison to those who develop clinical symptoms.

5. Figures 2, 3, 5, 6 and S1: When the p-value is marked as a line divided into two segments (e.g. Fig2, RBC, <2yrs) it is not clear whether there are significant differences between smear-negative samples and subclinical malaria and between subclinical and clinical malaria or whether there are also significant differences between smear-negative and clinical malaria. Based on my interpretation, I understand that there are no significant differences between smear-negative and clinical malaria, however, at first glance it seems that there are. Could the authors clarify this point? If there are significant differences between these groups, a clear definition in the legends and graphs on how the p-values are shown, should be provided.

Minor comments:

6. Line 60, 404 and 460: What does “adequate” means? The authors should be more concise describing a main conclusion.

7. Some references are missing along the manuscript (e.g. line 88, the authors should provide the reference of the Western population data; line 367, the authors may provide the reference of IL-6 role in iron homeostasis; line 417, the authors should provide the reference describing the significant role of hemolysis in asymptomatic malaria).

8. Line 264: Could de authors indicate if the difference in Hb level was calculated with the mean of the groups?

9. Line 281: To facilitate a better understanding of this section, I would recommend including the acronyms of the mature and immature reticulocytes that are shown in Fig S2.

10. Lines 330-331: The authors should clarify what the value in brackets means.

6. PLOS authors have the option to publish the peer review history of their article (what does this mean?). If published, this will include your full peer review and any attached files.

Reviewer #1: No

Reviewer #2: No

Reviewer #3: No

---

## [Author Response · Author response to Decision Letter 0]

29 Aug 2020

Additional analyses were done regarding the comment of Reviewer 1 for indepth analysis on the correlations of the biomarkers. These analyses were upload in a separate files as supporting documents to response to the reviewers comments.

---

## [Decision Letter · Decision Letter 1]

2 Oct 2020

PONE-D-20-20127R1

Red blood cell homeostasis in children and adults with and without asymptomatic malaria infection in Burkina Faso

PLOS ONE

Dear Dr. KABORE,

Thank you for submitting your manuscript for review to PLoS ONE. After careful consideration, we feel that your manuscript will likely be suitable for publication if it is revised to address a specific topic raised by the reviewer # 1. More specifically, previous suggestions made by the reviewer meant to make the manuscript more informative to a broad audience. Therefore additional analysis and data interpretation should be presented in the final version of the manuscript, even if only in supplementary materials, rather than just in the supporting file to the reviewers. For your guidance, a copy of the reviewers' comments was included below.

We look forward to receiving your revised manuscript.

Kind regards,

Luzia Helena Carvalho, Ph.D.

Academic Editor

PLOS ONE

Reviewers' comments:

Reviewer's Responses to Questions

**Comments to the Author**

1. If the authors have adequately addressed your comments raised in a previous round of review and you feel that this manuscript is now acceptable for publication, you may indicate that here to bypass the “Comments to the Author” section, enter your conflict of interest statement in the “Confidential to Editor” section, and submit your "Accept" recommendation.

Reviewer #1: All comments have been addressed

Reviewer #2: All comments have been addressed

Reviewer #3: All comments have been addressed

2. Is the manuscript technically sound, and do the data support the conclusions?

Reviewer #1: Yes

Reviewer #2: Yes

Reviewer #3: (No Response)

3. Has the statistical analysis been performed appropriately and rigorously? 

Reviewer #1: Yes

Reviewer #2: Yes

Reviewer #3: (No Response)

4. Have the authors made all data underlying the findings in their manuscript fully available?

Reviewer #1: No

Reviewer #2: Yes

Reviewer #3: (No Response)

5. Is the manuscript presented in an intelligible fashion and written in standard English?

Reviewer #1: Yes

Reviewer #2: Yes

Reviewer #3: (No Response)

6. Review Comments to the Author

Reviewer #1: 1) My question of “in-depth analysis on the correlations of the biomarkers to cluster and group biomarkers, and to draw some results and conclusion at the hyper level” on last version meant to make the manuscript more informative to all the audience instead of just to me, a reviewer. Therefore the relevant analysis results and interpretation should be presented in the manuscript, even if only in supplementary materials, rather than just in the supporting file to the reviewers.

Reviewer #2: (No Response)

Reviewer #3: (No Response)

7. PLOS authors have the option to publish the peer review history of their article (what does this mean?). If published, this will include your full peer review and any attached files.

Reviewer #1: No

Reviewer #2: No

Reviewer #3: No

---

## [Author Response · Author response to Decision Letter 1]

17 Oct 2020

Comments reviewer #1

Reviewer #1: 1) My question of “in-depth analysis on the correlations of the biomarkers to cluster and group biomarkers, and to draw some results and conclusion at the hyper level” on last version meant to make the manuscript more informative to all the audience instead of just to me, a reviewer. Therefore, the relevant analysis results and interpretation should be presented in the manuscript, even if only in supplementary materials, rather than just in the supporting file to the reviewers.

We thank reviewer for suggesion and we adapted our paper accordingly. See materials and method section: page 8; Lines 148-152 and results section: page 20-21; Lines 387-401. The corresponding figure and table are presented in the supporting materials.

---

## [Editor Report · Decision Letter 2]

4 Nov 2020

Red blood cell homeostasis in children and adults with and without asymptomatic malaria infection in Burkina Faso

PONE-D-20-20127R2

Dear Dr. KABORE,

We’re pleased to inform you that your manuscript has been judged scientifically suitable for publication and will be formally accepted for publication once it meets all outstanding technical requirements.

Kind regards,

Luzia Helena Carvalho, Ph.D.

Academic Editor

PLOS ONE
---

## [Editor Report · Acceptance letter]

16 Nov 2020

PONE-D-20-20127R2 

Red blood cell homeostasis in children and adults with and without asymptomatic malaria infection in Burkina Faso. 

Dear Dr. Kaboré:

I'm pleased to inform you that your manuscript has been deemed suitable for publication in PLOS ONE. Congratulations! Your manuscript is now with our production department. 

Kind regards, 

on behalf of

Dr. Luzia Helena Carvalho 

Academic Editor

PLOS ONE